# The Astragalus Membranaceus Herb Attenuates Leukemia by Inhibiting the FLI1 Oncogene and Enhancing Anti-Tumor Immunity

**DOI:** 10.3390/ijms252413426

**Published:** 2024-12-14

**Authors:** Kunlin Yu, Yao Tang, Chunlin Wang, Wuling Liu, Maoting Hu, Anling Hu, Yi Kuang, Eldad Zacksenhaus, Xue-Zhong Yu, Xiao Xiao, Yaacov Ben-David

**Affiliations:** 1State Key Laboratory for Functions and Applications of Medicinal Plants, Guizhou Medical University, Guiyang 561113, China; yukunlin511@163.com (K.Y.); tangy45@163.com (Y.T.); wangchunlin3323@163.com (C.W.); emmalao@163.com (W.L.); humaoting7@163.com (M.H.); anl_hu@163.com (A.H.); 18311855678@163.com (Y.K.); 2Natural Products Research Center of Guizhou Province, Guiyang 550014, China; 3Division of Advanced Diagnostics, Toronto General Research Institute, University Health Network, Toronto, ON M5G 1L7, Canada; eldad.zacksenhaus@utoronto.ca; 4Department of Microbiology & Immunology, Medical College of Wisconsin, Milwaukee, WI 53226, USA; xuyu@mcw.edu

**Keywords:** Astragalus membranaceus, traditional Chinese medicine, drug–protein interaction, FLI1, IDO1, DTO2, HDC, tumor microenvironment, leukemia inhibition

## Abstract

Astragalus membranaceus (AM) herb is a component of traditional Chinese medicine used to treat various cancers. Herein, we demonstrate a strong anti-leukemic effect of AM injected (Ai) into the mouse model of erythroleukemia induced by Friend virus. Chemical analysis combined with mass spectrometry of AM/Ai identified the compounds Betulinic acid, Kaempferol, Hederagenin, and formononetin, all major mediators of leukemia inhibition in culture and in vivo. Docking analysis demonstrated binding of these four compounds to FLI1, resulting in downregulation of its targets, induction of apoptosis, differentiation, and suppression of cell proliferation. Chemical composition analysis identified other compounds previously known having anti-tumor activity independent of the FLI1 blockade. Among these, Astragaloside-A (As-A) has marginal effect on cells in culture, but strongly inhibits leukemogenesis in vivo, likely through improvement of anti-tumor immunity. Indeed, both IDO1 and TDO2 were identified as targets of As-A, leading to suppression of tryptophane-mediated Kyn production and leukemia suppression. Moreover, As-A interacts with histamine decarboxylase (HDC), leading to suppression of anti-inflammatory genes TNF, IL1B/IL1A, TNFAIP3, and CXCR2, but not IL6. These results implicate HDC as a novel immune checkpoint mediator, induced in the tumor microenvironment to promote leukemia. Functional analysis of AM components may allow development of combination therapy with optimal anti-leukemia effect.

## 1. Introduction

Leukemias are a type of blood cancer originating in bone marrow with increased incidence and aggressiveness as a function of age. Acute myeloid leukemia is infrequent yet highly lethal [1]. Conventional Chemotherapy for leukemias is cytotoxic and relapse often renders incomplete patient recovery. Targeted therapy improves the clinical outcome, but drug resistance is a major obstacle [2]. Therefore, development of new effective therapies is urgently needed to combat this disease.

Traditional Chinese medicine (TCM) has been used for thousands of years to avert and manage human disease. TCM has also been considered as a complementary and alternative therapy [3]. As Western medicine generally focuses on effectively healing the patient’s symptoms, TCM theory is mainly based on the yin-yang and five element theories, the human body meridian systems, and the Zang Fu organ theory [4,5]. However, modern Chinese medicine has been developed to replace traditional methods that are based on herbs. The standard Chinese preparations now is in two popular forms, namely, extract (powders or granule) and smooth (like liquid or syrup) [6]. Astragalus membranaceus (AM) is a well-known and essential member of Chinese herbal medicine used to treat inflammation, neurodegenerative diseases, and other health issues [7,8]. AM is a flowering plant and its root has been frequently used in TCM. The major components of AM are polysaccharides, flavonoids, and saponins [7]. Beyond other diseases, AM exhibited anti-tumor activity by direct anti-proliferation or pro-apoptosis effect on various cell lines [9,10]. AM injection combined with chemotherapy improves short-term prognosis and clinical outcome in children with acute lymphocytic leukemia (ALL) [11]. Furthermore, AM ameliorates immunosuppression by activating M1 macrophages and T cell tumor-killing function in the tumor microenvironment (TME) [12]. AM also improves systemic immunity, which may help promote efficacy of chemotherapy and prevent metastasis [9].

Transcription factor Friend Leukemia Integration 1 (Fli-1) was originally discovered as an activated oncogene in erythroleukemias induced by Friend Murine Leukemia Virus (F-MuLV) [13,14,15]. Fli-1 is a member of the ETS-gene family of transcription factors capable of both activating and inactivating downstream target genes [16,17,18]. While EWS-FLI1 translocation is critical for development of Ewing’s sarcoma [19], FLI1 overexpression is observed in various human cancers including leukemias, breast cancer, and melanoma [16]. In addition to cancer, FLI1 plays critical roles in hematopoietic stem cell maintenance and development of various mature blood cells [16]. Specifically, FLI1 is involved in the development of angiogenesis and maturation of erythroid and megakaryocytic cells from Megakaryocytic Erythroid Progenitors [20,21]. Overexpression of FLI1 is associated with susceptibility to inflammation disorder systemic lupus erythematosus [22].

The Chinese drug administration approved AM as herbal injection and pills for the treatment of solid cancer, leukemia, and other health-related diseases in China. In this study, we demonstrate strong anti-leukemia activity of AM injection (Ai) in a mouse model of erythroleukemia induced by Friend virus. Consistent with the critical role of Fli-1 activation in leukemia and in this animal model, we found that AM/Ai strongly inhibits this ETS member in the culture of various leukemic cells leading to cell cycle arrest and apoptosis. Analysis of a large number of compounds from AM identified several agents that bind FLI1 and inhibit its activity. We also identified a compound, astragalus A (As-A), that strongly inhibited leukemogenesis in part by moderately blocking immune checkpoint IDO1/TDO2 and strongly suppressing inflammation within TME. These results implicate anti-FLI1 compounds and the immune modulator AS-A as important components for AM-mediated inhibition of leukemia.

## 2. Results

### 2.1. AM Injection Blocks Proliferation of Leukemia Cell Lines in Culture and Leukemogenesis in Mice

AM extracted from the Astragalus plant was previously shown to affect the proliferation of various cancer cell lines in vitro [9,10]. AM injection (Ai), mainly extracted from the traditional Chinese medicine Astragalus, was also used for the treatment of various diseases, including cancer [23]. Here, we show that Ai suppressed proliferation of erythroleukemia (HEL, K562) and monocytic (THP1) leukemic cell lines in culture with IC_50_ ranging from 60 to 100 mg/mL (Figure 1A). For control, the non-tumorigenic HEK293T and HL-7702 cell lines were treated with Ai with IC_50_ over 250 mg/mL. Since HEL cells exhibited higher sensitivity to Ai, we used this cell line in further analysis. Ai treatment inhibited proliferation of HEL cells in a dose-dependent manner (Figure 1B). Inhibition of cell growth was also shown by significant suppression of the proliferation marker ki67 (Appendix A). Inhibition of cell proliferation by Ai treatment was associated with the appearance of apoptotic cell death in a concentration-dependent manner (Figure 1C). FITC-annexin-V flow cytometry analysis also revealed increased percentage of apoptotic cells with higher concentrations of Ai treatment (Figure 1D). Moreover, Ai is shown to induce the cleavage of the apoptosis factor caspase-3 (Appendix A). Ai treatment reduced G1 and increased the G2 phase of the cell cycle in HEL cells in a dose-dependent manner (Figure 1E).

The mouse model of erythroleukemia induced by Friend Murine Leukemia Virus (F-MuLV) was used to determine the anti-leukemic activity of Ai in vivo. BALB/c mice (n= 7) inoculated at birth with F-MuLV were treated at five weeks post-virus infection with Ai (200 µL/mouse, is equivalent to 400 mg/mouse), every other day for two weeks. Mice were then monitored for development of leukemia and scarified before they became morbid. Treatment with Ai significantly inhibited the rate of leukemia development when compared with the control vehicle only (saline) group (Figure 1F). The Ai-treated mice exhibited a smaller spleen weight than the control group due to leukemia suppression (Figure 1G). As hematocrit strongly dropped in the control group, this level was elevated in the Ai-treated mice (Figure 1H). Flow cytometry analysis of spleens from the leukemic mice revealed that treatment with Ai led to higher number of terminally differentiated TER119^+^ and lower number of CD71^+^ erythroid progenitor cells (Appendix A). In bone marrow, the percentages of TER 119^+^ and CD71^+^ cells were elevated in Ai-treated leukemic mice compared to the control group (Appendix A). These results demonstrate a strong anti-neoplastic effect of Ai in culture and leukemic mice.

### 2.2. Inhibition of Leukemogenesis by Ai Is Mediated in Part Through Suppression of FLI1

Induction of erythroleukemia is mediated through retroviral insertional activation of *fli*-1 [13,14]. We therefore examined whether FLI1 function was impaired by Ai treatment, leading to leukemia inhibition. Western blot analysis revealed that human FLI1 protein expression in HEL cells was significantly downregulated by Ai in a dose-dependent manner (Figure 2A). The microRNA, MiR145, controls FLI1 protein expression in cancer cells [24,25,26,27,28], and is a downstream target of FLI1, which negatively regulates its expression in an autoregulatory loop [28]. Accordingly, Ai treatment inhibited FLI1 function in a dose-dependent manner resulting in upregulation of miR145 transcription (Figure 2A,B). Interestingly, FLI1 transcription was also inhibited directly by Ai treatment (Figure 2C). Downregulation of FLI1 in HEL cells resulted in higher expression of the erythroid differentiation markers CD235a and CD71 (Figure 2D). These results demonstrate that Ai suppresses growth at least in part by inhibiting FLI1 activity. Consistent with this, IC_50_ of Ai is significantly higher in lentivurus-FLI1 knockdown (shFLI1-HEL) versus scrambled-control cells (Figure 2E,F).

### 2.3. Identification of AM Compounds Exhibiting Leukemia Inhibitory Activity and Anti-FLI1 Function

The composition of Astragalus membranaceus (AM) extract has been extensively studied and 87 compounds were identified by Traditional Chinese Medicine Systems Pharmacology (TCMSP) (https://old.tcmsp-e.com accessed on 12 April 2023) (Appendix A). Among these, 20 compounds are considered active based on oral benefit (OB) [higher than 30%] and drug-like properties (DL) [higher than 0.18] (Figure 3A,B). Interestingly, many active compounds in the AM list show anti-cancer activity including Betulinic Acid [29], Kaempferol [30], Quercetin [31], Formononetin [32], Hederagenin [33], Isoflavanone [34], Bidendate [35], and calysosin [36]. To identify which of these compounds exhibit anti-FLI1 activity, molecular docking analysis was performed against FLI1 protein (Figure 3B). Several compounds were identified with a binding energy above −7 kcal/mol. Betulinic Acid (Ba), Quercetin, and Kaempferol were among molecules with highest binding affinity (above −7.6 kcal/mol) to FLI1. Betulinic acid (Figure 4A) exhibited strong affinity to the FLI1 protein binding domain (Figure 4B), causing downregulation of FLI1 protein in HEL cells in a dose-dependent manner (Figure 4C). In Cellular and Thermal Shift Assay (CETSA), at higher temperature Ba increased FLI1 protein stability, further confirming this affinity (Appendix A). We previously demonstrated that drug-mediated inhibition of FLI1 triggers upregulation of MiR145, responsible for FLI1 protein downregulation [28]. Suppression of FLI1 by Ba caused inactivation of FLI1, which subsequently induced its downstream target MiR145, reducing FLI1 protein expression. Indeed, Ba treatment of HEL cells led to a dose-dependent activation of MiR145 (Figure 4D), associated with FLI1 protein downregulation (Figure 4C). However, Ba treatment of HEL cells resulted in no significant changes in the level of FLI1 mRNA expression (Figure 4E).

Treatment of HEL cells with Ba resulted in strong downregulation of cell proliferation in a dose- and time-dependent manner (Figure 4F). In the animal model of erythroleukemia, Ba strongly inhibited the progression of leukemia (Figure 4G), associated with smaller leukemic spleen weight (Figure 4H) and higher hematocrit (Figure 4I), when compared to control DMSO-treated mice. These results implicated Ba as a specific anti-FLI1 compound with the ability to inhibit leukemias expressing high levels of FLI1.

### 2.4. Binding of Kaempferol to FLI1 Inhibits Leukemic Growth in Culture and Suppresses Erythroleukemogenesis

The Kaempferol (Ka) compound exerts a strong anti-cancer activity against several types of cancers [30]. Like Ba, Ka also interacts with FLI1 at high affinity (−7.6 kcal/mol; Figure 5A,B), leading to dose-dependent downregulation of FLI1 protein in HEL cells (Figure 5C). The FLI1 binding by Ka was also confirmed using CETSA (Appendix A). Downregulation of FLI1 protein was associated with dose-dependent upregulation of MiR145, leading to FLI1 protein degradation (Figure 5D). However, some level of FLI1 mRNA downregulation was also detected by RT-qPCR of HEL cells treated with Ka (Figure 5E). Treatment of HEL cells with Ka resulted in strong downregulation of cell proliferation in a dose- and time-dependent manner (Figure 5F). This is consistent with the result that IC_50_ level of Ka is significantly higher in shFLI1-HEL versus scrambled-control cells (Figure 5G). In HEL cells, treatment with Ka increased erythroid differentiation associated with upregulation of the human erythroid markers CD235a and CD71 (Appendix A). As FLI1 is known to negatively block GATA1 to induce erythroid differentiation [22], the level of GATA1 was also increased at both transcription (Appendix A) and protein levels (Appendix A) in HEL cells treated with Ka (15, 30 μM). In our animal model of erythroleukemia, Ka significantly delayed erythroleukemia progression (Figure 5H). By flow cytometry, Ka treatment increased the number of erythroid TER 119^+^ cells and decreased the level of CD71^+^ progenitors in the spleen (Appendix A). The percentages of TER 119^+^ and CD71^+^ cells were increased in bone marrow of Ka-treated mice (Appendix A). These results demonstrate both Ba and Ka as potent anti-FLI1 and anti-leukemia agents in AM extracts.

### 2.5. Identification of Compounds Responsible for Anti-Leukemic Activity of Ai

Next, we subjected Ai extracts to mass spectrometry (MS) to identify its chemical composition and identified over 170 compounds (Appendix A). Table 1 summarizes the identity of agents with the highest binding energy to FLI1, as determined by molecular docking analysis and their anti-proliferative activity in HEL cells. Most of these compounds display binding affinity to FLI1 greater than −6.5 kcal/mol. Some of the listed compounds are not commercially available (see Table 1, marked as bold). Among these compounds, hederagenin (He) and formononetin (Fn) have a binding energy of −7.6 and −6.8 kcal/mol to FLI1, respectively (Figure 6A,B,F,G). The affinity of both He and Fn to FLI1 was further confirmed using CTESA (Appendix A). Both compounds display strong inhibitory activity on HEL cells in culture (Figure 6C,H), and downregulated FLI1 protein and upregulated MiR145 (Figure 6D,E,I,J). Both He and Fn were previously shown to have anti-neoplastic activity [32,37] that may be mediated through suppression of the FLI1 activity.

Astragaloside A (As-A) (Figure 7A), known as the most active component of AM [38,39,40], exhibited a binding energy of −7.4 kcal/mol to FLI1 (Table 1 and Figure 7B), but weakly inhibited the proliferation of HEL cells in culture with IC_50_ of 80µM. As-A also failed to downregulate the expression of FLI1 protein when administrated to HEL cells in culture (Figure 7C). Accordingly, the expression of MiR145 was only mildly affected by this compound (Figure 7D). Interestingly, despite weak anti-leukemic activity in culture, in the mouse model of erythroleukemia, As-A strongly inhibited leukemia progression (Figure 7E). These results raised the possibility that when combined with other Ai compounds, As-A may suppress leukemia through strengthening the immune system. This possibility is consistent with the health benefit of Astragalus purported to enhance the immune system and reduce inflammation [12,41,42,43,44].

As the immune checkpoint blockade for CTLA4, PD1 and ligand PDL1 disrupts negative immune regulatory checkpoints to release pre-existing anti-tumor immune responses [45,46]. However, the majority of patients receiving these blockade therapies do not show a clear clinical response [46]. Targeting additional immune checkpoints, co-stimulatory receptors, and/or co-inhibitory receptors that control T cell function and other immune-components are therefore of great interest [47]. Indeed, additional immune checkpoint blockades were recently identified, including LAG3, TIM3, IDO1, TDO2, TNFRSF4, and TIGIT [46,47]. Docking analysis identified a strong affinity of As-A to IDO1 and TDO2 with binding energies of −9.5 and −9.1 kcal/mol, respectively (Appendix A). IDO1 and TDO2 are the rate-limiting enzymes necessary for the conversion of Trp to Kyn [48]. Kyn presence can attenuate cytotoxic T-cell activation and result in a pro-tumor microenvironment [49]. Expression of both IDO1 and TDO2 are known to be induced in cells by IFN-γ [50]. While the expression of IFN-γ in HEL cells is very low, treatment with this cytokine and As-A did not significantly affect its mRNA expression (Figure 7F). As expected, the expression of both IDO1 and TDO2 mRNA was strongly increased by IFN-γ (Figure 7G,H). The IDO1/DTO2 induction by IFN-γ was lightly but significantly reduced in presence of both IFN-γ + As-A (Figure 7G,H). As-A alone exhibited no significant effect on the expression of IFN-γ and TDO2, but slightly reduced the IDO1 level relative to the control (Figure 7F–H). While IFN-γ treatment of HEL cells increased Kyn, treatment of HEL cells with both IFNγ plus As-A slightly reduced tryptophan metabolism (Figure 7I). Interestingly, a previous study demonstrated IFN-γ-induced kynurenine activation and induction of inflammatory response by FLI1 in nasopharyngeal carcinoma [51]. In our leukemic cells, however, the expression of IDO1/TDO2 was not significantly affected by the anti-FLI1 compound. Ba-induced Kyn metabolic activity slightly increased in Ba + IFN-γ treated cells, indicating a marginal role for FLI1 in this enzymatic process in leukemia (Appendix A–C). The modest inhibition of the Trp-Kyn pathway by As-A suggests involvement of another immune modulation pathway by this compound.

We previously demonstrated the importance of the enzyme histamine decarboxylase (HDC), which catalyzes decarboxylation of histidine to generate histamine, on biological processes including inflammation, allergy, asthma, and leukemia [52]. While HDC inhibition had no effect on cell proliferation in culture, this enzyme is thought to alter the TME through the activation of inflammatory factors, leading to leukemia progression. Indeed, Diacerein, a potent inhibitor of the HDC and inflammatory factors, strongly suppresses leukemia progression [52]. We found that As-A displayed a strong binding affinity (−8.4 kcal/mol) to HDC (Appendix A). Treatment of HEL cells with As-A resulted in downregulation of HDC (Figure 7J) as well as its downstream inflammatory effector genes TNF (Figure 7K), IL1B (Figure 7L), CXCR2, IL1A, and TNFAIP3, but not IL6 (Appendix A). These results suggest that As-A acts as a potent inhibitor of HDC, which functions as a potentially new immune checkpoint modulator to affect inflammation within the TME to promote leukemia.

## 3. Discussion

The Astragalus membranaceus plant is commonly used in many herbal formulations to treat a wide variety of diseases and disorders. Herein, an AM injection formula is shown to strongly prevent leukemia progression induced in mice by Friend virus. We identified at least four compounds within AM/Ai with anti-FLI1 activity, resulting in suppression of leukemia. AM extracts contain additional compounds with known anti-leukemia activity, independent of FLI1. Astragaloside A (As-A), the known active component of AM, is shown herein to exhibit a very weak inhibitory activity on leukemic cells in culture but to strongly suppress leukemic progression in mice through promotion of anti-tumor immunity, reducing inflammation within the TME. These results provide the underlying mechanism of AM-induced leukemia inhibition that may ultimately allow the development of personalized combination therapy of human leukemias expressing high levels of FLI1 [28].

FLI1 is a powerful oncogene that promotes the progression of diverse cancers [16]. In addition, this ETS gene is involved in the etiology of several diseases [17]. Targeting this transcription factor is therefore of great interest for the treatment of malignancies and diseases associated with aberrant FLI1 expression. Our analysis of AM chemical composition described herein identified four compounds (Ba, Ka, He, and Fn) exhibiting strong anti-FLI1 activity, associated with the induction of differentiation, apoptosis, and inhibition of leukemia proliferation. The compounds He and Fn, only detected in Ai, are commonly used in the clinic as an injection extraction of astragalus. All of these compounds were previously studied as anti-cancer agents [29,30,32,33], although the underlying mechanism and their targets are unknown. AM extract also contains other compounds including Isoflavanone, Quercertin, Bidendate, Calysosin and others, previously known to display anti-cancer activity [31,34,35,36]. It is therefore the combinational effect of these compounds that is responsible for the anti-leukemic activity of Ai on culture cells and in vivo model of Friend erythroleukemia. As Ai comprises over 170 compounds, we cannot exclude the possibility that some of these molecules accelerate cell proliferation or exert some toxicity that may compromise the optimal anti-leukemia activity of the extract. Therefore, analyzing of the anti-neoplastic activity of each compound within AM/Ai is necessary to develop a personal drug formulation with the best outcome for cancer therapy.

As-A compounds surprisingly demonstrated marginal anti-proliferative activity on leukemic cells in culture, but strongly blocked leukemia in diseased mice. The As-A effect in vivo was even stronger than anti-FLI1 compounds studied in this report. AM indeed was previously suspected to exert some of its anti-neoplastic ability by inducing an anti-cancer immune response [9,12]. Several recent studies even demonstrated that As-A targets immune-checkpoint proteins IDO1 and TDO2 [44,53]. Both IDO1 and TDO2 are responsible for the conversion of Trp to Kyn that block anti-cancer activity of cytotoxic T cells within TME [49,50]. Consistent with this, our docking analysis revealed a strong binding affinity of As-A to both IDO1 and TDO2, but this compound did not change the expression of these proteins. Instead, we detected moderate but significant inhibition of Kyn activity in leukemic cells treated with As-A. It is likely that As-A inhibits leukemia in part through suppression of the immune checkpoint modulators IDO1 and TDO2. As Kyn activity was recently reported to be induced by FLI1 in nasopharyngeal carcinoma [51], this process was not affected by the FLI1 inhibitor Ba, indicating different regulation of the inflammatory process by this transcription factor in leukemia.

In a previous study, we demonstrated a critical role that the enzyme histamine decarboxylase (HDC) has as an immune modulator in promoting leukemia [52]. HDC is regulated by FLI1 and other transcription factors to control histamine production. HDC inhibition by specific inhibitors such as Epigallocatechin (EGC) and diacerein exhibited marginal effect on cell proliferation in culture, but strong anti-inflammatory effect in TME, resulting in potent tumor inhibition [52]. As-A is shown herein to bind to HDC and exhibit a similar effect on leukemia development resulting in inhibition of several anti-inflammatory genes including TNF, IL1A/B, TNFA1P3, and CXCR2, but not IL6. These results suggest that As-A likely acts as a potent anti-inflammatory modulator capable of altering TME and immune cells to block leukemia progression. Indeed, macrophages and activated T-cells are known to express HDC (Human Protein Atlas, http://www.proteinatlas.org, Single cell sequencing, 11 December 2024). HDC may therefore function as a new immune checkpoint regulator of leukemia progression. As FLI1 also regulates HDC, some of its anti-leukemia activity is likely mediated through HDC suppression. Interestingly, the level of leukemia suppression by anti-FLI1 inhibitors is slightly less than that seen in mice treated with As-A. This may be because of other pro-immunity functions of As-A through blocking IDO1/TDO2 or other checkpoints. It is also possible that while leukemia suppression activity of anti-FLI1 compounds is robust, inhibition of FLI1 may activate some pro-tumorigenic pathway(s) that counteracts some of its anti-neoplastic function. Future studies are needed to further investigate these possibilities and develop a combinational therapy using the AM compounds with optimal anti-neoplastic activity to be used for treatment of leukemia.

In addition to cancer, AM is used to treat lupus erythematosus allergy/asthma, and this is consistent with the role of FLI1 in these inflammatory diseases [21,54,55,56,57,58]. This raises the possibility that AM administration may exert these therapeutic benefits through FLI1 as well. This observation also raises the possibility that other inflammatory diseases treated by AM may also be mediated through the function of FLI1, a notion that warrants further investigation in future studies.

Overall, this study demonstrates for the first time the important contribution of at least four AM components in blocking FLI1 function, leading to leukemia inhibition. Other AM components exhibit anti-cancer activity although through other mechanisms. Among these, we identified As-A as an important mediator of immunity, in part through the inhibition of IOD1/TDO2 checkpoints and suppression of pro-inflammatory HDC in leukemia progression (Figure 8). HDC suppression may then represent a new marker of immune checkpoint activation, leading to suppression of leukemogenesis. The outcome of this study may allow us to generate a combinational therapy consisting of compounds exerting optimal anti-leukemia activity to be used for the treatment of human blood cancer, including AML.

## 4. Materials and Methods

### 4.1. Cell Lines

The human microplasma-tested erythroleukemia cell lines (HEL, K562, HL60), monocyte leukemia cell line THP1, the human embryonic kidney 293 (HEK293T), and human normal liver HL-7702 cells were previously obtained from ATCC (Manassas, VA, USA) and maintained mycoplasma-free. These cell lines were cultured and maintained in Dulbecco’s Modified Eagle Medium supplemented with 5% fetal bovine serum (HyClone, GE Healthcare, Chicago, IL, USA).

### 4.2. shRNA Expression

The shRNA expression plasmid (12 mg) and packaging plasmid psPAX2 (6 mg), pMD2.G (12 mg) (Didier Trono, Lausanne, Switzerland, Addgene plasmid # 12259 and # 12260) were mixed and transfected into HEK293T cells using Lipofectamine^®^2000 (Thermo Fisher Scientific, #11668019, Waltham, MA, USA). The supernatants were collected and used to infect HEL cells. Lentivirus of scrambled vector was generated with a similar strategy. The positive cells after transduction were selected and cultured for 24 h using RPMI-1640 medium containing puromycin (5 mg/mL; Solarbio, Beijing, China).

### 4.3. RT-qPCR Analysis

Total RNA was extracted from the culture of HEL cells (4 × 10^5^) using TRIzol^®^ (Thermo Fisher Scientific, Inc.) and the manufacturer’s recommended protocol. RNA concentrations were measured using a NanoDrop 2000 spectrophotometer (Thermo Scientific Fisher, Inc.). To generate cDNA, a reverse transcription reaction was performed using the PrimeScript RT Reagent kit (Takara Biotechnology Co., Ltd., Shiga, Japan). RT-qPCR was performed using FastStart Universal SYBR Green Master (Roche Diagnostics GmbH, Mannheim, Germany) and the Step One Plus Real-time PCR system (Applied Biosystems, Foster City, CA, USA; Thermo Fisher Scientific, Inc.). The expression was normalized to GAPDH. The 2^−ΔΔCq^ method was used for relative quantification. RNA extraction, cDNA synthesis, and RT-qPCR were all performed according to the manufacturer protocols. The primer sequences are reported in Appendix A. A total of three biological triplicates were used for all RT-qPCRs, each in triplicates (n = 3).

### 4.4. Western Blotting

Cells were collected and protein was extracted using RIPA lysis buffer (Solarbio, Seattle, WA, USA) and phenylmethanesulfonyl fluoride (PMSF). After lysis and protein density determination using a BCA protein assay kit (Solarbio, USA), 50 μg samples were loaded on 10% acrylamide gel and transferred onto the PVDF membrane. The membrane was blocked with TBS buffer containing 5% skimmed milk for 1.5 h at room temperature. Polyclonal rabbit antibodies for FLI1 (cat. no. ab133485) [dilution 1:1000] and GATA1 (ab181544) [dilution 1:2000] were purchased from Abcam, Cambridge, UK; Anti-Caspase 3/CASP3 (p17) (PB9188) [dilution 1:500] was purchased from BOSTER, San Mateo, CA, USA; GAPDH (cat. no. AB-P-R 001) [dilution 1:1000] from Hangzhou Goodhere Biotechnology Co., Ltd., Hangzhou, China; and secondary antibodies (Anti-rabbit IgG (H + L) (DyLight™ 800 4X PEG Conjugate)) from CST Biological Reagents Co., Ltd., Danvers, MA, USA (cat. no. 5151S) [dilution 1:30,000]. The antibodies were diluted with TBS buffer containing 3% BSA, the primary antibodies were incubated overnight at 4 °C, and the secondary antibodies were incubated for 1.5 h at room temperature. The Odyssey Imaging System (LI-COR Biosciences, Lincoln, NE, USA) was used for Western blot protein imaging, and the protein density was determined using the software (Odyssey CLX Image Studio 3.1) that accompanied the system.

### 4.5. Molecular Docking

The 3D structure of all compounds was generated by KingDraw software v5.0 and saved in mol2 format. The crystal structure of FLI1(ID:5E8G) was downloaded from RCSB Protein Data Bank (RCSB PDB) and then dehydrated and hydrogenated using AutoDockTools-1.5.6. The molecular docking between human FLI1 protein and compounds was conducted using AutoDockTools-1.5.6. The molecular docking results were imported and visualized by PyMOL 2.4.1.

### 4.6. Assays of Cellular Differentiation

Immunofluorescence staining was conducted to detect erythroid cells, as previously described (21). In brief, 1 × 10^5^ cells were stained with APC-conjugated antibodies for 40 min at 4 °C. Cells were then washed twice and resuspended in 200 µL PBS and used for flow analysis. The following primary antibodies were used: human CD71-APC (cat. no. 551374) and human CD235a-APC (cat. no. 551336) purchased from BD Biosciences, Franklin Lakes, NJ, USA. Flow cytometry was performed using a NovoCyte flow cytometer and Novoexpress software 1.6.2 (ACEC Biosciences Inc., Chengdu, China).

The gating strategies were used as follows: FSC-A/SSC-A plots were used to separate live cells from debris. Erythroid cells were differentiated using a scatter/anti-CD71+ and a scatter/anti-CD235a+ gate from the unstained control, respectively. Count/anti-CD71+ and count/anti-CD235a+ in histograms present the expression of these markers.

### 4.7. Cell Viability Assays

The logarithmically grown HEL, K562, THP-1, HEK293T, HL7702, and HL60 cells were seeded in a 96-well culture plate at a density of 8000 cells per well and treated with different concentrations of Ai (0, 50, 100, 200, 400 mg/mL), Ba (0, 5, 10, 20, 40 µM), Ka (0, 10, 20, 40, 80 µM), He (0, 10, 20, 40, 80 µM), Fn (0, 10, 20, 40, 80 µM), As-A (0, 10, 20, 40, 80 µM) or 0.1% DMSO (control). The concentration of Ai was obtained based on the amount of active drug in the injection solution (2 g/mL) provided by manufacturer. After 24 h, 48 h, 72 h, and 96 h treatment, the cells were added to 10 µL of MTT reagent for 4 h, and then added to 100 µL of triple cleavage fluid (SDS 10 g, isobutanol 5 mL, and 10M HCl 0.1 mL, dissolved with ddH2O into 100 mL solution). After using a microscope to observe the completely dissolved crystal, the cells’ optical absorbance value was measured at 570 nm using an enzyme-labeling instrument (BioTek Instruments, Inc., Vermont, USA).

### 4.8. Cellular Thermal Shift Assay (CETSA)

When the HEL cell density reached 90–100%, it was treated with Ba (20 μM), Ka (30 μM), He (20 μM), Fn (20 μM), or DMSO for a duration of 2 h. Subsequently, the cells were collected and resuspended in lysis buffer containing PMSF. The cell suspension was divided into 7 equal portions and subjected to heat treatment at various temperatures (40, 43, 45, 47, 49, 52, and 55 °C) for a period of 3 min followed by cooling at 4 °C for another 3 min. After undergoing two cycles of freezing and thawing using liquid nitrogen and heating at 25 °C, respectively, the dissolved protein present in the supernatant was obtained through centrifugation at a speed of 12,000 rpm for 15 min at 4 °C. Finally, the obtained protein was transferred into new centrifuge microtubes to facilitate detection of FLI1 protein expression via Western blots.

### 4.9. Apoptosis and Cell Cycle Analyses

For the apoptosis detection assay, the logarithmically grown HEL cells were treated with Ai (0, 50, 100, and 200 mg/mL) for 48 h. Subsequently, the cells were collected and rinsed three times with pre-cooled PBS. Based on the instructions for the Annexin V-FITC apoptosis detection kit (Becton, Dickinson, Franklin Lakes, NJ, USA), the cells were resuspended in 50 µL of 1× binding buffer and stained for 15 min. Then, the apoptosis rate was determined using flow cytometry (ACEA NovoCyte, San Diego, CA, USA). For cell cycle analysis, HEL cells were treated with the different concentrations of Ai (0, 50, 100, and 200 mg/mL) for 24 h. Cells were collected, fixed with pre-cooled 70% ethanol, and kept at 4 °C for 4 h before being transferred to −20 °C overnight. Cells were stained with 0.5 mL of PI (propidium iodide, 50 µg/mL), RNase (5 µg/mL) inhibitor, and Triton × 100 (0.5 µg/mL) for 30 min at room temperature in the dark and then analyzed by flow cytometry.

### 4.10. Immunofluorescence Staining

The HEL cells were collected for the determination of Ki67 by the Ki67 Cell Proliferation Assay Kit (IF, Red, Rabbit mAb) (C2312S, Beyotime, Shanghai, China) following the manufacturer’s instructions. HEL cells were washed with PBS and cytospun onto adhesion microscope slides (CITOTEST), immunofluorescence stained, and then fixed with 4% paraformaldehyde for 20 min under room temperature; cells were washed 3 times with washing liquid and then blocked for 30 min with blocking solution. The cells were rinsed with PBS and incubated overnight at 4 °C in Ki67 rabbit monoclonal antibody. The cells were washed 3 times with washing liquid, then incubated with anti-rabbit CY3 for 1 h, and 4′,6-diamino-2-phe nylindole (DAPI) was used for the nucleus counterstaining. Finally, the cells were visualized using a fluorescence microscope at 40× magnification.

### 4.11. Animal Experiments

The NIH3T3 cells expressing the F-MuLV clone 57 vector were used to induce leukemia. BALB/c mice (male and females; Tongxin, Chongqing, China), aged one day, were administered a solitary intraperitoneal injection of F-MuLV. Then, infected neonates were weaned at 4 weeks and randomly divided into 2 groups: control group (Con) and Ai, Ba, Ka, or As-A group (Ai: 200 µL/mice, Ba: 5 mg/kg, Ka: 10 mg/kg, or As-A: 15 mg/kg). After 5 weeks of virus infection, mice (n = 7) were treated with Ai, Ba, Ka, or As-A intraperitoneally every other day for 2 weeks. The control group was given saline or DMSO. Under the supervision of experienced staff, the leukemia mice were sacrificed humanely by cervical dislocation. Blood was taken from the heart and the volume ratio measured. The spleens and humerus of the mice were removed. Part of the spleen tissue was ground and bone marrow was washed out from the humerus by saline, and then prepared as a single-cell suspension, which was stained with CD71 and TER 119 antibodies for 30 min, respectively. The expression of markers on the spleen and bone marrow cell surfaces of mice was analyzed by flow cytometry.

### 4.12. Enzyme-Linked Immunosorbent Assay

HEL cells were plated into 3 cm dishes 4 h. Consequently, cells were pretreated with As-A (0 and 10 µM) and Ba (0, 15 and 20 µM) for 2 h, following co-incubation with IFN-γ (40 ng/mL) for 24 h. The medium was collected for the determination of Kyn by the Elisa Kit (COIBO, BIO, Beijing, China) following the manufacturer’s instructions.

### 4.13. Ethics Statement

All animal experiments were approved by the ethics committee of State Key Laboratory for Functions and Applications of Medicinal Plants, Guizhou Medical University, with the approval number 2200956.

### 4.14. Statistical Analysis

All data are presented as the mean ± standard deviation (SD) from at least three independent experiments. Statistical analysis was performed using Student’s *t*-test or one-way ANOVA in GraphPad Prism 8.0 (GraphPad Software, La Jolla, CA, USA) and SPSS 26.0 (IBM Corporation, Armonk, NY, USA). *p* < 0.05 was considered statistically significant.

## Figures and Tables

**Figure 1 ijms-25-13426-f001:**
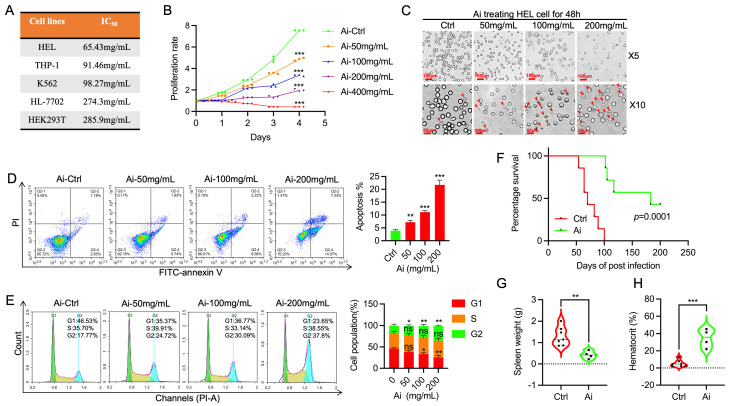
Astragalus injection (Ai) inhibits leukemia survival and proliferation in culture and blocks leukemogenesis in mice. (**A**) IC_50_ analysis of Ai treatment for the indicated leukemia cell lines in culture. (**B**) Proliferation rate of Ai treatment on HEL cells using the indicated concentration of drug. (**C**) Microscopic images of HEL cells treated with the indicated concentration of Ai. (**D**) Apoptosis index of HEL cells treated with the indicated concentration of Ai. Average of three experiments is shown in right panel. (**E**) Cell cycle index of HEL cells treated with the indicated concentration of Ai treatment. Average of three experiments is shown in the right pane. (**F**) Survival rate of leukemic mice treated with saline or Ai. (**G**,**H**) Spleen weight (**G**) and hematocrit (**H**) of control and Ai-treated mice at 90 days after virus inoculation. Data are expressed as mean ± SD, n = 3 independent experiments. *** *p* < 0.001, ** *p* < 0.01 and * *p* < 0.05, ns: no significant differences.

**Figure 2 ijms-25-13426-f002:**
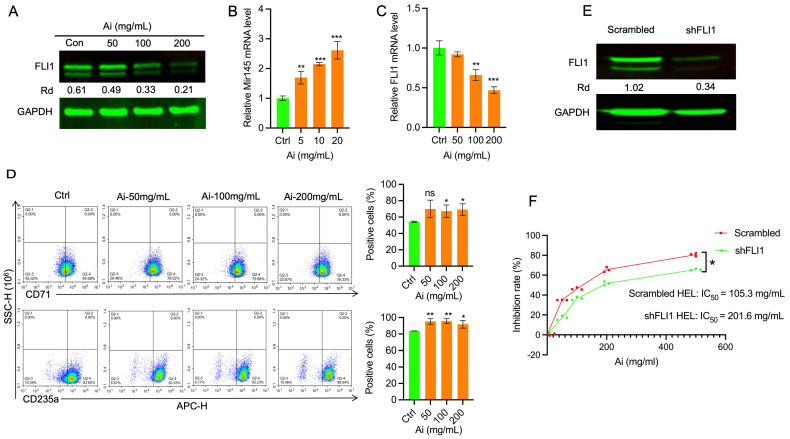
Ai treatment inhibits FLI1 function leading to leukemia suppression. (**A**) Expression of FLI1 protein in HEL cells treated with the indicated concentration of Ai. Relative density (Rd) of the band is indicated. (**B**) Expression of MiR145 in HEL cells treated with the indicated concentration of Ai, by RT-qPCR. (**C**) Expression of FLI1 mRNA in HEL cells treated with the indicated concentration of Ai, by RT-qPCR. (**D**) Expression of erythroid markers CD71 and CD235a in HEL cell treated with Ai or vehicle control, as determined by flow cytometry. Relative percentage of three experiments is shown in the right panel. (**E**) Knockdown of FLI1 in HEL cells using shFLI1, as determined by Western blotting. (**F**) Increase IC_50_ by Ai treatment in shFLI1-HEL versus scrambled-control cells. Data are expressed as mean ± SD, n = 3 independent experiments. *** *p* < 0.001, ** *p* < 0.01 and * *p* < 0.05, ns: no significant differences.

**Figure 3 ijms-25-13426-f003:**
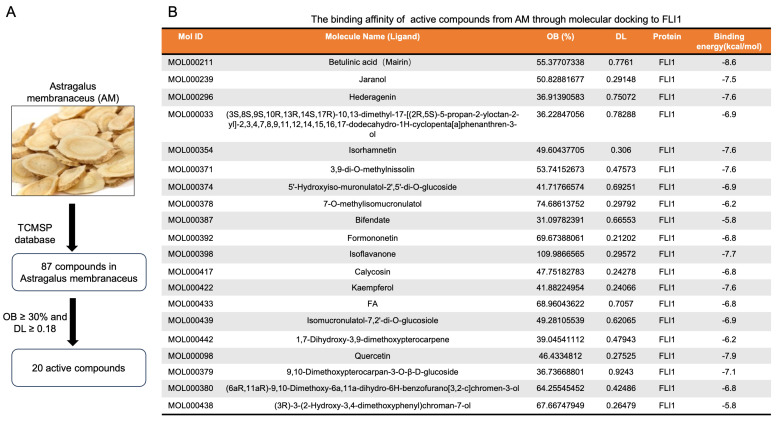
AM contains several compounds with potent anti-FLI1 activity. (**A**) The diagram of AM extraction toward identification of active compounds from the published TCMSP database list. The active compounds are based on criteria such as oral benefit (OB) [higher than 30%] and drug-like property (DL) [higher than 0.18]. (**B**) List of 20 active compounds and their anti-FLI1 activity based on their affinity to bind FLI1. The binding energy to FLI1 for each compound (kcal/mol) is provided.

**Figure 4 ijms-25-13426-f004:**
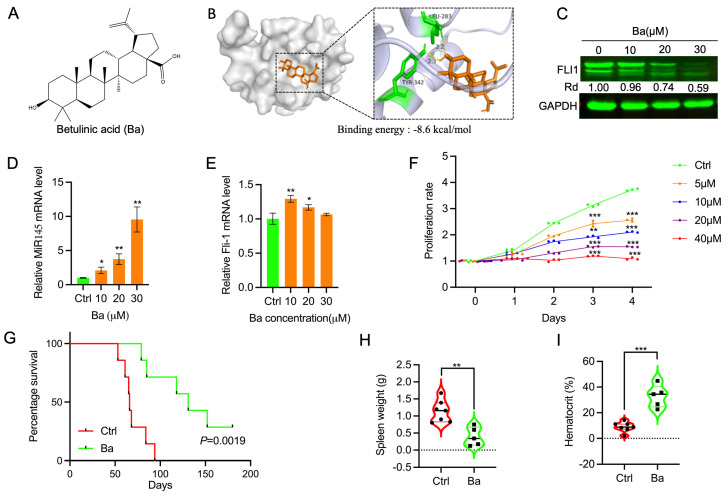
Betulinic acid (Ba) binds to FLI1 and inhibits HEL cell proliferation and suppresses leukemogenesis. (**A**) The chemical structure of Ba. (**B**) Three-dimensional (left) and two-dimensional (right) interaction of Ba to FLI1 and its binding energy. (**C**) Ba downregulates FLI1 protein in a dose-dependent manner. (**D**) Ba induces MiR145 transcription in a dose-dependent manner. (**E**) Effect of Ba on FLI1 transcription, determined by RT-qPCR. (**F**) Ba inhibits HEL cell proliferation dose dependently in culture. (**G**) Ba blocks leukemogenesis in the mouse model of erythroleukemia induced by F-MuLV. (H-I) Spleen weight (**H**) and hematocrit (**I**) of leukemic mice treated with Ba. Data are expressed as mean ± SD, n = 3 independent experiments. *** *p* < 0.001, ** *p* < 0.01 and * *p* < 0.05.

**Figure 5 ijms-25-13426-f005:**
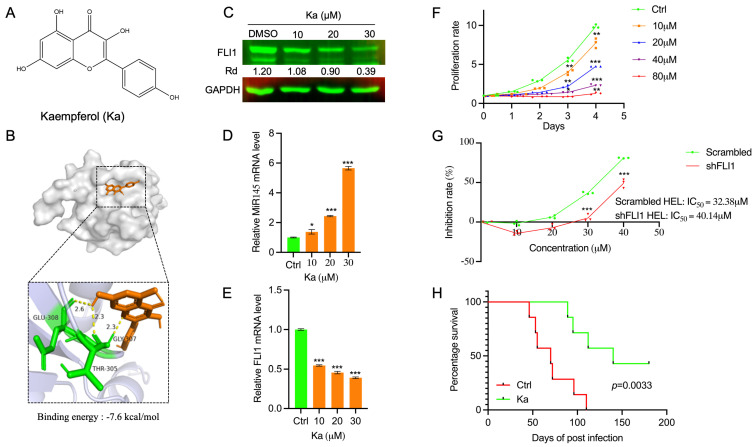
Kaempferol (Ka) blocks FLI1 function, causing inhibition of HEL cell proliferation and suppression of erythroleukemogenesis. (**A**) The chemical structure of Ka. (**B**) Three-dimensional (top) and two-dimensional (bottom) structure of binding of Ka to FLI1 protein with its binding affinity (kcal/mol). (**C**) Ka treatment of HEL cells downregulates FLI1 protein expression, by Western blotting. (**D**) Ka increases the transcription of MiR145 in HEL cells, in a dose-dependent manner. (**E**) Ka treatment downregulates FLI1 mRNA expression in HEL cells. (**F**) Ka treatment of HEL cells suppresses cell proliferation in a dose-dependent manner. (**G**) Ka exhibits a higher IC_50_ in shFLI1-HEL cells versus scrambled-control cells. (**H**) Ka significantly suppresses erythroleukemogenesis in vivo. Data are expressed as mean ± SD, n = 3 independent experiments. *** *p* < 0.001, ** *p* < 0.01 and * *p* < 0.05.

**Figure 6 ijms-25-13426-f006:**
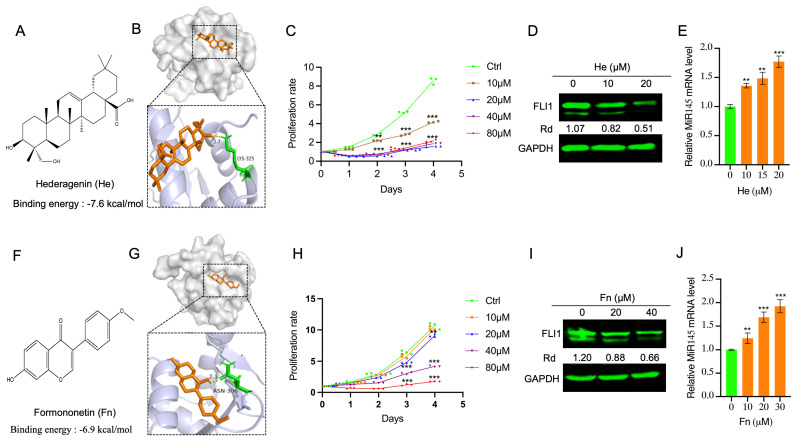
Hederagenin (He) and Formononectin (Fn) inhibits FLI1 to block HEL cell proliferation and suppress erythroleukemia cell proliferation. (**A**,**F**) Molecular structure of He (**A**) and Fn (**F**). (**B**,**G**) Three- and two-dimensional structure of interaction of He (**B**) and Fn (**G**) to FLI1 with their binding energy (kcal/mol). (**C**,**H**) Inhibition of HEL cell proliferation by He (**C**) and Fn (**H**) in culture at the indicated doses. (**D**,**I**) Downregulation of FLI1 protein by He (**D**) and Hn (**I**) at the indicated doses, determined by Western blot. (**E**,**J**) Induction of MiR145 by He (**E**) and Fn (**J**), by RTqPCR. Data are expressed as mean ± SD, n = 3 independent experiments. *** *p* < 0.001 and ** *p* < 0.01.

**Figure 7 ijms-25-13426-f007:**
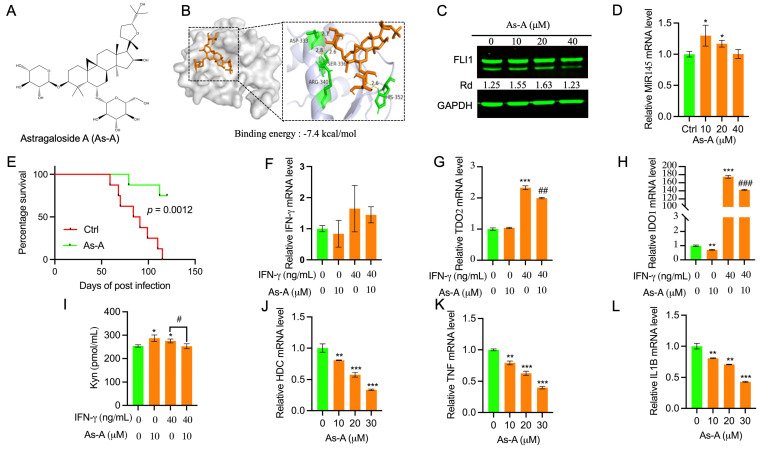
Astragaloside A (As-A) inhibits leukemia by altering immune checkpoints. (**A**) Chemical structure of As-A. (**B**) Three- and two-dimensional structure of As-A with its binding energy to FLI1, determined by molecular docking. (**C**) Effect of As-A on FLI1 expression, as determined by Western blotting. (**D**) Effect of As-A on MiR145 expression using the indicated doses. (**E**) As-A (15 mg/kg) inhibits leukemia in the mouse model of erythroleukemia induced by F-MuLV. (**F**–**H**) Expression of the IFN-γ (**F**), TDO2 (**G**), and IDO1 (**H**) genes in HEL cells treated with or without IFN-γ and As-A, as determined by RT-qPCR. (**I**) Kyn activity in HEL cells treated with or without IFNγ and As-A, as determined by Elisa. (**J**–**L**) Expression of the HDC (**J**), TNF (**K**), and IL1B (**L**) in HEL cells treated for 24 h with the indicated doses of As-A, determined by RT-qPCR. Data are expressed as mean ± SD, n = 3 independent experiments. *** *p* < 0.001, ** *p* < 0.01 and * *p* < 0.05 versus Ctrl groups; ^###^
*p* < 0.001, ^##^
*p* < 0.01 and ^#^
*p* < 0.05 versus IFN-γ groups.

**Figure 8 ijms-25-13426-f008:**
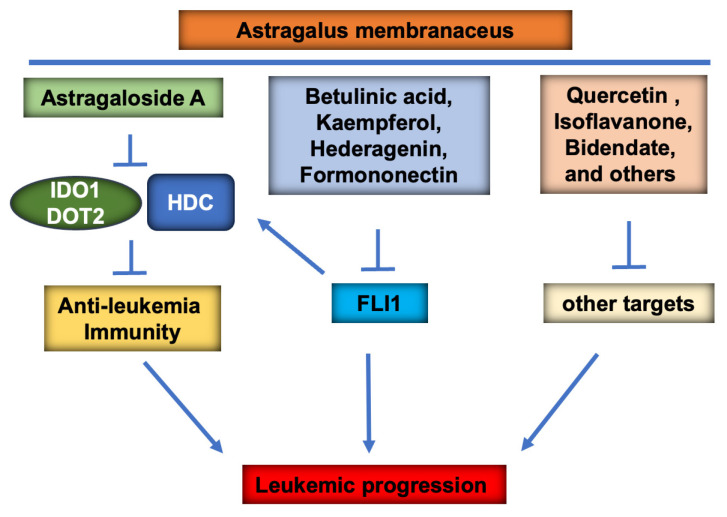
Graphical image. The mode of leukemia inhibition by Astragalus membranaceus (AM). AM contains a large number of compounds exhibiting various functions to confer anti-cancer activity. At least four compounds, Ba, Ka, He, and Fn, exert their anti-cancer activity through inhibition of FLI1 function. Other compounds also inhibit leukemia through other mechanisms. While the As-A compound displays a marginal direct effect on leukemia cells in culture, it blocks immunity against leukemia cells by blocking the function of the checkpoint proteins IDO1/DOT2 and pro-inflammatory protein HDC within the tumor microenvironment. The combine effect of these compounds represents the optimal inhibitory response of AM on leukemic cells.

**Table 1 ijms-25-13426-t001:** The binding affinity of compounds from Ai to FLI1.

Protein	Molecule Name (Ligand)	Binding Energy (kcal/mol )	Effect on FLI1	Drug Efficacy
FLI1	Daucosterol	−7.7	Poor	Poor HEL cell inhibitory effect
FLI1	Astragaloside A	−7.4	Poor	Poor HEL cell inhibitory effect
FLI1	Astragaloside II	−7.6	No detection	No HEL cell inhibitory effect
FLI1	Acetytastragaloside	−7.7	No detection	No detection
FLI1	Astragaloside III	−8.4	Poor	Poor HEL cell inhibitory effect
FLI1	Astrachrysoside A	−8.3	No detection	No detection
FLI1	Mangiferonic acid	−7.9	No detection	No detection
FLI1	(8S,11R,13S,14S,17S)-11-(1,3-benzodioxol-5-yl)-17-hydroxy-13-methyl-17-prop-1-ynyl-1,2,6,7,8,11,12,14,15,16-decahydrocyclopenta[a]phenanthren-3-one	−9.28	Poor	Poor HEL cell inhibitory effect
FLI1	**1-ethynyl-1-hydroxy-11a-methyl-hexadecahydro-1H-cyclopenta[a]phenanthren-7-one**	−8.69	No detection	No detection
FLI1	**1-{7-hydroxy-9a,11a-dimethyl-1H,2H,3H,3aH,3bH,4H,6H,7H,8H,9H,9aH,9bH,10H,11H,11aH-cyclopenta[a]phenanthren -1-yl}ethyl 3-(4-hydroxyphenyl)prop-2-enoate**	−8.68	No detection	No detection
FLI1	Hederagenin	−7.6	Moderate	Moderate HEL cell inhibitory effect
FLI1	Formononetin	−6.9	Moderate	Moderate HEL cell inhibitory effect
FLI1	Corticosterone	−8.64	No detection	No detection
FLI1	Tupisteroide A	−8.2	No detection	No detection
FLI1	**1-(furan-3-yl)-1,8-dihydroxy-3b,6,6,9a,11a-pentamethyl-1H,2H,3bH,4H,6H,7H,9aH,9bH,10H,11H,11aH-cyclopenta[a]phenanthrene-4,7-dione**	−8.08	No detection	No detection
FLI1	**3,10-dihydroxy-4,7,11,12b,14a-pentamethyl-2,8,8a,9,10,11,12,12a,12b,13,14,14a-dodecahydropicene-2,9-dione**	−8.5	No detection	No detection
FLI1	**11a-methyl-7-oxo-1H,2H,3H,3aH,3bH,4H,5H,7H,8H,9H,9aH,9bH,10H,11H,11aH-cyclopenta[a]phenanthren-1-yl acetate**	−8.36	No detection	No detection
FLI1	**3-methyl-6-(1-{4,7,10-trihydroxy-3a,6,6,9a,11a-pentamethyl-1H,2H,3H,3aH,4H,5H,5aH,6H,7H,8H,9H,9aH,10H,11H,11aH-cyclopenta[a]phenanthren-1-yl}ethyl)oxan-2-one**	−8.4	No detection	No detection
FLI1	**1-acetyl-9a,11a-dimethyl-hexadecahydro-1H-cyclopenta[a]phenanthren-7-yl 2-(4-benzoylphenyl)acetate**	−9.04	No detection	No detection

Unavailable drugs are marked as bold.

## Data Availability

The data presented in this study are available on request from the corresponding author.

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
