# Peer review of "The Astragalus Membranaceus Herb Attenuates Leukemia by Inhibiting the FLI1 Oncogene and Enhancing Anti-Tumor Immunity"

_ijms, 2024, doi:10.3390/ijms252413426_

Round 1
Reviewer 1 Report
Comments and Suggestions for Authors
In general, the manuscript is clear, and the introduction is exhaustive; the experimental design is linear and well-structured, and the methodologies described are precise and faithfully reproduce the experiments performed. The data are consistent to demonstrate the hypothesis, and the results are presented clearly. The discussion is linearly whit the appropriate references.
A suggestion to improve the consistency of the data on cellular proliferation inhibition and apoptotic cell death induction by AM extract treatment, would be appropriate add a protein quantification of some markers such as Ki67 and Caspase3 to Figure1 Panel C.
Considering the evidence that AM is an extract considered active based on oral benefit, do you have tested also an oral formulation of the compound? Does the pharmacokinetic of the AM and of the AM compounds permits to opt for oral administration maintaining the same effect?
Page 5 line 186-192 text needs to be justified.
Page 6line 206-226 text needs to be justified and formatted.
Discussion text needs to be justified.
Author Response
请参阅附件。

Reviewer 2 Report
Comments and Suggestions for Authors
Here are my comments for manuscript #3338275:
Comments:
1. Authors used "Ai injection" on manuscript. It is acceptable for mice study but for cell line study, please use "Ai treatment". Actually, "Ai treatment" can be used for both cell lines and mice studies.
2. Figure 1A: need the data from HEK293 which is not cancer cell.
3. Figure 1C: The cells look very different after Ai treatment. Please describe and discuss the morphology change.
4. There is a need to add apoptosis study in Figure 1.
5. The current study investigates several compounds. Is there any additive or synergistic effect of those compounds? Since all compounds are from Astragalus membranaceus, study the additive and synergistic effect is necessary.
6. Please list all abbreviations.
Round 2
Reviewer 2 Report
Comments and Suggestions for Authors
No more comments.